# Local Production of Acute Phase Proteins: A Defense Reaction of Cancer Cells to Injury with Focus on Fibrinogen

**DOI:** 10.3390/ijms25063435

**Published:** 2024-03-19

**Authors:** Péter Hamar

**Affiliations:** Institute of Translational Medicine, Semmelweis University Budapest, Tűzoltó u 37–47, 1094 Budapest, Hungary; hamar.peter@semmelweis.hu

**Keywords:** local acute phase response (lAPR) cancer, tumor microenvironment (TME), fibrinogen (FN), modulated electrohyperthermia (mEHT)

## Abstract

This review is intended to demonstrate that the local production of acute phase proteins (termed local acute phase response (lAPR)) and especially fibrin/fibrinogen (FN) is a defense mechanism of cancer cells to therapy, and inhibition of the lAPR can augment the effectiveness of cancer therapy. Previously we detected a lAPR accompanying tumor cell death during the treatment of triple-negative breast cancer (TNBC) with modulated electro-hyperthermia (mEHT) in mice. We observed a similar lAPR in in hypoxic mouse kidneys. In both models, production of FN chains was predominant among the locally produced acute phase proteins. The production and extracellular release of FN into the tumor microenvironment is a known method of self-defense in tumor cells. We propose that the lAPR is a new, novel cellular defense mechanism like the heat shock response (HSR). In this review, we demonstrate a potential synergism between FN inhibition and mEHT in cancer treatment, suggesting that the effectiveness of mEHT and chemotherapy can be enhanced by inhibiting the HSR and/or the lAPR. Non-anticoagulant inhibition of FN offers potential new therapeutic options for cancer treatment.

## 1. Introduction of the Topic

The heat shock response (HSR) is a well-described general and ancient stress response of cells to various types of stress. The acute phase response (APR) is a part of the immune response to infections and tissue damage. Proteins whose plasma concentration is changed by at least 25% in response to pro-inflammatory stimuli are termed acute phase proteins (APPs) [1]. They have a role in restoring homeostasis after inflammation [2]. The generally held concept is that APPs are produced in the liver; these are triggered mainly by the inflammatory interleukin-6 (IL-6) (as well as IL-1, IL-8 and TNF-α), and secreted into the blood. However, APPs are also synthesized in other organs. Thus, APPs are a part of local defense responses and can contribute to repair mechanisms [3]. We demonstrated by multiplex analysis a complex local production of several APPs in different tissues upon stress, such as electromagnetic heating of tumor cells [4], acute hypoxic [5] or chronic fibrotic injury of the kidney [6]. Thus, we propose that the local APP production is an organized, ancient cellular stress response, like the much-better-characterized HSR [7]. The local APR (lAPR) can be an important mechanism to protect cells from stress similarly to the HSR [8]. While APPs are traditionally known for their systemic roles in the acute phase response, there is increasing recognition that they can also exert canonical and non-canonical functions in local tissue defense. Although, besides hepatocytes, inflammatory cells are thought to be the key sources of systemic acute phase proteins, other cells such as stromal cells, adipocytes, endothelial cells [9] and cancer cells [10] have been described to produce acute phase proteins. APPs can modulate inflammatory signaling pathways [11]. For example, they may interact with cell surface receptors or soluble mediators to regulate the intensity and duration of the inflammatory response [9]. By fine-tuning inflammation, APPs can help prevent excessive tissue damage while still promoting effective host defense. Furthermore, APPs can also modulate immune cell function and communication within the tissue. They may influence the activation, differentiation, and function of immune cells, such as macrophages [12,13], neutrophils [14], and dendritic cells [15]. By shaping the immune response at the local level, APPs help coordinate effective host defense while minimizing collateral tissue damage. Finally, APPs play roles in tissue repair and remodeling processes beyond their traditional roles in inflammation. For instance, some acute phase proteins, such as fibronectin and tenascin, are involved in cell adhesion, migration, and extracellular matrix remodeling [16]. By promoting tissue integrity and repair, these proteins contribute to the restoration of tissue homeostasis following injury or infection. 

An association between acute-phase proteins (APPs) and cancer has long been established and many cancerous cells produce APPs. The biological significance of the lAPR in cancer, and the benefit of cancer cells from these proteins remains largely unknown. Recent data revealed that some APPs, including α1-antitrypsin, are able to enhance cancer cell resistance to anticancer drug-induced apoptosis [10].

Inhibition of the HSR can support cancer therapies as demonstrated by us [8] and others [17,18]. Similarly to the HSR inhibition, inhibition of the local APR may be beneficial in cancer treatment [4,19]. 

## 2. Background: Description of the Model Used in the Reviewed Experiments

### 2.1. Breast Cancer (BC)

Breast cancer (BC) is one of the most frequent cancer types among women worldwide. Triple-negative breast cancer (TNBC) is a highly aggressive BC type with very poor survival due to the lack of targeted therapy. The most used mouse TNBC models utilize cell lines derived from mouse mammary carcinoma cell line 410.4, isolated from a single spontaneous tumor in Balb/c mice. The cell lines 4T1 and 4T07 are the most aggressive and invasive sub-clones [19]. Inoculation of these syngeneic cell lines creates isografts in Balb/c mice. Thus, immune mechanisms can be investigated under conditions very similar to those of human TNBC [8].

### 2.2. Modulated Electro-Hyperthermia (mEHT)

Modulated electro-hyperthermia (mEHT) is a newly emerging adjuvant cancer treatment used in human oncology, with strong cancer inhibitory effect in monotherapy in our mouse model. During mEHT, a focused electromagnetic field (EMF) is generated within the tumor by applying capacitive radiofrequency. Selective energy absorption by the tumor is the consequence of its elevated oxidative glycolysis (Warburg effect) and conductivity. The EMF induces cell death by thermal and non-thermal effects. Capacitive energy delivery and frequency modulation enable the application of non-thermal effects. We have developed the rodent mEHT device to enable accurate, reproducible, standard, and effective treatment of TNBC in the inguinal region of mice [8]. Selective energy absorption enabled to warm up the tumor by +2.5 °C—a great advantage over former hyperthermia methods, where the temperature difference between the tumor and surrounding tissues was only +1 °C. Thus, the thermal and non-thermal effects amplify each, other leading to effective tumor cell killing. The device is also useful for local delivery of therapeutic agents by using thermo-sensitive liposomes (TSL). 

## 3. Experimental Evidence Demonstrating Local Acute Phase Protein and Fibrinogen (FN) Production in Cancer Treated with Modulated Electro-Hyperthermia 

We have demonstrated that mEHT-induced tumor cell death is located in the core of treated tumors. The necrotic area was cleaved caspase-3 (cC3) positive, suggesting apoptotic cell death. We observed a strong HSP-70 staining in the treated tumors around the dead area, suggesting induction of the heat-shock response (HSR) by mEHT [8].

Multiplex analysis of mEHT-treated TNBC [4,8] and hypoxic [5] or fibrotic [6] kidney samples allowed us to identify a proteomic response dominated by APPs as detailed below (Figure 1). Comparison of the raw data from our four papers analyzing tissue/cell stress with multiplex methods revealed that several APPs were upregulated in more than one model. Furthermore, real-time polymerase chain reaction (RT-PCR) detection of mRNA and cell culture studies confirmed that the APPs were produced locally by the tumor or renal cells, confirming that APPs were produced by the stressed cells themselves and not by the liver. 

Contrary to most previous studies, where multiplex RNA (next-generation sequencing, NGS) results were verified by few, individual data (using, e.g., qPCR), we verified our NGS data by another multiplex method: Nano-String (NS) nCounter^®^ (NanoString Technologies, Seattle, WA, USA), using the bar-coding of each RNA in the sample. We selected 134 target genes from the NGS data to create a custom NanoString gene panel. Nano-String verified 104 target genes with the same direction of change as detected by NGS (Figure 1C). Furthermore, we performed mass spectrometry (MS) as a multiplex method to verify gene expression at the protein level.

### The Local Acute Phase Response (lAPR)

APPs are plasma proteins synthesized in the liver whose concentrations increase (or decrease) by 25% or more during inflammation [20]. This definition is based on the electrophoretic separation of plasma proteins, which demonstrates elevation of the alpha1 and alpha2 fractions [21]. APPs and the APR in general are stably conserved in evolution [22]. These proteins play a crucial role in the innate immune response and the body’s defense mechanisms, and serve to protect tissues during infection, inflammation and/or tissue damage [23]. The production of APP’s maintains homeostasis and tissue repair, but also represents a central component of the organism’s defense strategy, especially in the context of innate immunity [21]. 

In contrast to the systemic APR, where the APPs are produced by the liver, proteins of the lAPR are produced in tissues in response to injury or inflammation. Unlike APPs produced in the liver, which circulate throughout the bloodstream, local APPs are primarily synthesized at the site of injury or inflammation. They help regulate the inflammatory response and promote tissue repair.

The background of the identified APPs and their known contribution to cancer progression are summarized in a recent review [11]. The lAPR identified in our screens are all major APPs: all 3 FN chains (FGA, FGB, FGG), haptoglobin (HP), complement component 4B (C4B) and several protease inhibitors: inter-alpha-trypsin inhibitor (I alpha I) family members (light chain (AMBP) and heavy chains: ITIH-1,3,4), clade A (extracellular) serpins (especially A1 (alpha-1 antitrypsin) A3B (alpha-1 antichymotrypsin), and A3K (kallikrein inhibitor). 

The following is true for all of these proteins: (1) they are APPs; (2) the main site of their synthesis is the hepatocyte, where besides basal expression, inflammation upregulates their synthesis during APR; (3) extra-hepatic production has been demonstrated already; (4) they are well preserved throughout evolution (ancient proteins) found in many lower species; (5) they are involved in cellular stress response (defensive). 

Indeed, several previous publications support our observation that local APP production is an ancient defense mechanism. Although these scarce and isolated findings demonstrate local production of some APPs, but the coordinated, complex local APR has not been described before. A proteomic analysis of the urine revealed APPs (FGA, FGG, HP, ITI4, SERPINA1) [14] as biomarkers for prostate cancer without further analysis. 

## 4. Fibrin(ogen) (FN)

### 4.1. The Structure of FN

FN is a complex protein composed of two sets of disulfide-bridged polypeptide chains: 2 Aα (alpha), 2 Bβ (beta), and 2 γ (gamma) chains [24]. Each chain has distinct roles in the process of blood clotting and FN formation: Aα chains play a crucial role in the polymerization process of fibrinogen. They contain binding sites that interact with platelets and other molecules involved in clot formation. Additionally, Aα chains contribute to the mechanical properties of the fibrin clot, influencing its stability and structure. Bβ chains also participate in polymerization of fibrinogen, contributing to the formation of the fibrin clot. They contain sites that are involved in interactions between fibrin molecules, helping to link them together during clot formation. Gamma chains are important for the overall structure of fibrinogen. They contribute to the stability and integrity of the molecule. Although not directly involved in fibrin clot formation, they play a critical role in maintaining the overall structure of fibrinogen, which influences its function in clotting. The fibrinogen-like domain is found in all three types of fibrinogen chains, and contributes to the assembly of fibrinogen molecules into a larger, functional fibrin polymer with specific binding sites for fibrinogen, allowing fibrinogen molecules to connect and form a mesh-like structure.

The leukocyte integrin, αMβ2 (Mac-1), is a high affinity receptor for fibrin(ogen) on stimulated monocytes and neutrophils that is important for fibrin(ogen)–leukocyte interactions that contributes to the inflammatory response. This domain binds the CD11b/CD18 complement receptor-3 (CR-3, Mac-1) leading to macrophage activation [25]. The Mac-1-binding site within the fibrinogen D domain contains two peptide sequences, γ190–202 and γ377–395, designated as P1 and P2 [24].

### 4.2. The Physiological (Canonical and Non-Canonical) Functions of Fibrin(ogen) (FN) in the Blood and in the Extracellular Matrix (ECM)

Fibrinogen is a glycoprotein that is involved in blood clotting and is synthesized in the liver. However, it can accumulate in injured tissues and contribute to local inflammation and tissue repair. Fibrinogens are soluble proteins in the blood. Fibrinogen chains (Aα, Bβ, γ) are present in the human blood at concentrations of 1.5–4 g/L, thus being one of the most abundant plasma proteins (7–8%) after albumin [26]. 

The physiological functions of FN are: in addition to providing a scaffold for blood clotting, FN is important for the assembly of (extracellular) matrices to enhance host defense [27]. Upon passing out of the bloodstream, fibrinogens become unsolvable and precipitate in the ECM. The fibrin mesh formed in the ECM serves as a provisional matrix essential for tissue repair [9]. Increased production of matrix molecules such as fibrinogen and protease inhibitors during the APR serve as a general mechanism to promote tissue repair (Table 1).

Proteins such as fibrinogen (fibrinogen-related proteins) comprise a collection of extracellular molecules, each containing a conserved fibrinogen-like globe: a common and ancient domain that has evolved to play key roles in tissue repair. The ancestral function of fibrinogen domain-containing molecules was defense [28]. Fibrinogen-related proteins are upregulated during tissue repair [29]. Thus, cancer cells stressed by hyperthermia or chemotherapy can produce fibrinogen locally as a mechanism of defense and repair.

Over time, as organisms evolved and developed more complex systems, fibrinogen and its related molecules may have acquired additional functions beyond basic clotting and defense. In vertebrates, fibrinogen-related proteins play roles not only in hemostasis but also in inflammation, immune regulation, and tissue repair [29]. Specific, non-coagulatory functions of fibrinogen include stimulation of angiogenesis, keratinocyte and fibroblast migration and proliferation, matrix remodeling and the recruitment and activation of immune cells [29]. Understanding the ancestral functions of fibrinogen domain-containing molecules provides insights into their diverse roles in modern organisms, which include not only coagulation but also aspects of immunity and tissue homeostasis.

### 4.3. Documented Roles of Fibrinogens in Non-Hematologic Disease

While fibrinogen is primarily known for its crucial role in blood clotting and coagulation, the involvement of fibrinogen in non-hematologic diseases has been widely documented. 

#### 4.3.1. Fibrin(ogen) Plays a Role in Inflammatory Conditions

However, precipitated fibrin is also inflammatory. Fibrin- and interferon-induced genes are inflammatory [30]. FN has been implicated in pathological processes including renal diseases and cancer. FN-deficient mice (Fg^−/−^) [31] were protected from endotoxemia [32], renal fibrosis [33] and ischemia-reperfusion injury [34]. In fibrinogen KO mice nerve injury was reduced. Fibrin precipitated in the ECM activates local inflammatory cells (e.g., microglia in the brain) but may also induce plaque formation directly in neurodegenerative diseases. 

Fibrinogen and its downstream degradation products (D-dimer and other fibrin degradation products) are widely used as diagnostic markers in inflammatory conditions, including COVID-19 [28]. 

Fibrinogen has been associated with the development and progression of atherosclerosis and elevated fibrinogen levels were demonstrated to be a predictor of cardiovascular disease in the Framingham study [35].

In patients with rheumatoid arthritis serum fibrinogen was elevated and correlated with markers of inflammation [36]. Also, in inflammatory bowel disease, increased fibrinogen levels were demonstrated to be a predictive biomarker of disease activity [37].

Elevated fibrinogen levels are associated with insulin resistance and may contribute to the increased cardiovascular risk in individuals with type 2 diabetes [38]. Diabetes is associated with increased plasma fibrinogen especially in patients with albuminuria [38]. Fibrinogen levels are often elevated in obesity. This may contribute to the pro-inflammatory state associated with obesity and its complications. Obese patients show increased fibrinogen levels and associated with insulin resistance and metabolic syndrome [39].

FN appears extracellularly in the brain in neurodegenerative diseases such as Alzheimer’s dementia (AD) or progressive multiple sclerosis (MS) [40,41], where fibrinogen deposition is linked to plaque formation [42]. Increased vascular permeability is associated with worse prognosis in these diseases. It has been demonstrated that fibrinogen is causative in the neurodegenerative process [40]. Furthermore, fibrinogen has been documented to be deposited in different brain diseases as a neuropathological hallmark. The causative role of FN is supported by the observation that FN KO mice are rescued from cognitive impairment (Alzheimer’s disease).

Elevated fibrinogen levels are linked to inflammation, exacerbations, severity, and mortality in chronic obstructive pulmonary disease [43].

Taken together FN and its extracellular deposition has a proinflammatory role in several diseases.

#### 4.3.2. Documented Roles of Fibrin(ogen) in Cancer Progression

Molecular mechanisms induced by extracellular fibrin deposition and their relevance in cancer progression are summarized in Table 1 and Figure 2A. Coagulation factors have been linked with malignancy for over 100 years, and high plasma fibrinogen has been associated with cancer progression. Fibrin(ogen) (FN) may enter the tumor stroma from the circulation as evidenced by FN deposition in the vicinity of blood vessels in tumors [44]. The therapeutic removal of such FN deposition has been demonstrated to enhance therapy delivery to the cancer [45].

However, FN can be produced by cancer cells [46], such as BC cells [27]. The locally produced FN binds to and surrounds cancer cells, forming a protective structure. Furthermore, by interacting with endothelial cells, FN contributes to the extravasation of cancer cells [26]. A hallmark of breast carcinoma is the local synthesis and deposition of fibrinogen (precipitation without cleavage by thrombin to fibrin) [47]. FN in the ECM augments the innate immune response to tissue injury or cancer [47]. FN deposition is a predominant component in breast tumor stroma [47]. In fibrinogen KO mice, the carcinogenesis gene-network was downregulated [30]. In our studies, all three FN chains were within the top upregulated genes on all three multiplex screens in TNBC and renal tissue. Our multiplex data do not distinguish between fibrinogen and fibrin and do not indicate the involvement of thrombin or other coagulation-related factors. 

From the identified panel of APPs in our studies, most data on cancer progression are available on FN. As reviewed by us previously [48], the data are conflicting regarding beneficial or deleterious effects of FN on cancer progression. FN is important for the assembly of (extracellular) matrices to enhance host defense [27]. High plasma FN has been associated with cancer progression. FN produced by BC cells [27] binds to and surrounds cancer cells, forms a protective structure, and contributes to extravasation of BC cells [26]. A hallmark of BC is the local synthesis and deposition of FN (precipitation without conversion to fibrin) making FN a predominant component of BC stroma [47]. Furthermore, FN augments the innate immune response to cancer [47]. 

**Table 1 ijms-25-03435-t001:** Molecular mechanisms induced by extracellular fibrin deposition and their relevance in cancer progression. Abbreviations: FN: fibrin, ECM: extracellular matrix, PS: phosphatydil-serine, *Fgα^−/−^*: fibrinogen alpha chain knockout, NK-cell: natural killer cell, αMβ2: integrin receptor, EC: endothelial cell.

FN Function/Molecular Mechanism	FN in Cancer Progression	Experimental Evidence	Reference
FN filtered from the circulation (FN deposition in the vicinity of blood vessels in tumors)	Compact stroma hinders therapy delivery	Removal of the compact stroma enhances therapy delivery	[44,45,49]
	Enhancement of cancer cell extravasation	Lactadherin—by competing with FN for PS binding sites may delay tumor progression	[26,50]
FN stimulates ECM formation	FN scaffold in the TME → adhesion, migration, invasion and metastasis of cancer cells	FN is a predominant component of the BC stroma. FN KO mice (*Fgα^−/−^*) were protected against hematogenous metastasis	[26,27,51,52]
	FN binds to + surrounds cancer cells → protective structure from treatments and immune system	FN interaction with platelets via β3-integrins facilitates the protection of tumor cells from NK-cell cytotoxicity, permitting escape from host immune surveillance	[26,27,30]
FN is important for the assembly of ECM to enhance host defense	FN contributes to the tumor favorable TME	In fibrinogen KO mice, the carcinogenesis gene-network was downregulated	[27,53]
	FN supports cancer cell growth, survival and proliferation	Primary colon cancer development diminished in *Fgα^−/−^* mice	[54,55]
FN net: growth factor reservoir	FN contributes to the proinflammatory TME to favor tumor progression	Interaction with αMβ2 induce leukocyte adhesion to ECs + proinflammatory cytokine production	[47,55]

#### 4.3.3. Fibrinolysis

Fibrinolysis pathways and their drugability is demonstrated on Figure 2B. Besides FN upregulation, fibrinolysis inhibitor serpins (A1, A3) were also upregulated in our multiplex screens. Thus, removing FN from the tumor microenvironment (TME) by plasmin may have a synergistic potential with anti-cancer therapies including mEHT. On the other hand, plasmin may have pro-cancer effects by releasing cancer-related growth factors and promoting angiogenesis [56]. Thus, modulating FN production may have a therapeutic potential.

Furthermore, serine protease inhibitors (extracellular clade A serpins) are produced in the liver and different cancer cells. Serpins participate in fibrinolysis, invasion, and migration and thus metastasis of cancer cells, and their expression is associated with cancer progression and poor prognosis [57], supporting their role in cancer cell protection. Besides FN upregulation, fibrinolysis inhibitors Serpin A3 (alpha-1-Antichymotrypsin) and alpha-2-macroglobulin were also upregulated in both the tumor and the renal samples. Thus, these data by us and others support a central role of local fibrinogen production and deposition in the TME of breast cancer.

**Figure 2 ijms-25-03435-f002:**
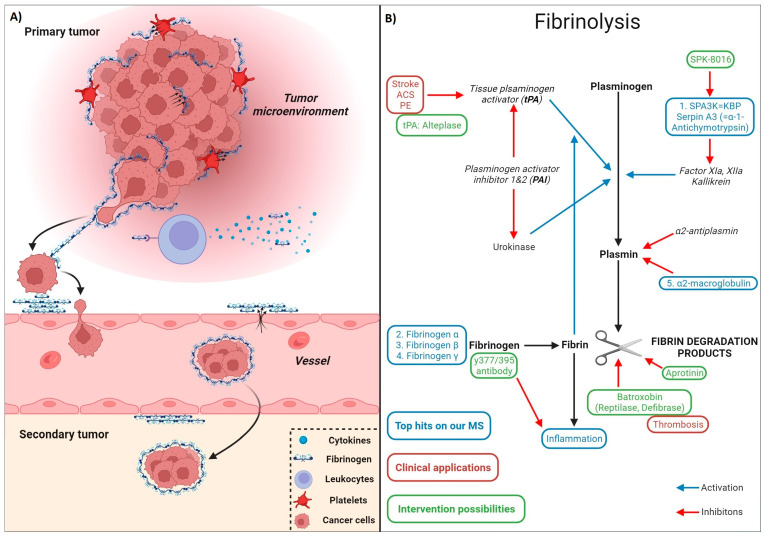
(**A**) Fibrin(ogen) surrounding tumor cells produced by the cancer cells themselves or platelets in the TME aids the formation of protective tumor microenvironment. The deposited fibrin may stimulate proinflammatory processes in the TME by recruiting and activating leucocytes. Fibrin filtered from permeable blood vessels and deposited around them aid the vascular entrance of potentially metastatic cells [26]. (**B**) Drugability of the fibrinolysis pathways. Top upregulated proteins (blue) in our mouse model were related to fibrin(ogen) deposition and degradation. (tPA: tissue plasminogen activator [58], PAI: plasminogen activator inhibitor, SPA3K: serine protease inhibitor (Serpin)-A3 = alpha-1-anitchymotrypsin). Current possibilities for intervention (green labels) and diseases where these interventions are used clinically (green letters) are indicated. (ACS: acute coronary syndrome, PE: pulmonary embolism, tPA: tissue plasminogen activator [59]: direct thrombin inhibitor (dabigatran), SPK: Spa3K inhibitor developed by Spark Therapeutics [60]. The figures were created with Biorender.com.

#### 4.3.4. Fibrin Induces a Tumor-Favorable Microenvironment (TME)

A TME includes blood vessels, immune cells, fibroblasts, and extracellular matrix components. FN contributes to the tumor-favorable TME as follows: (1) Stimulating angiogenesis that increases oxygen and nutrient supply to the tumor [53]. Tumor hypoxia-induced VEGF production by the cancer cells increases vascular permeability and consequent fibrin leakage which directly stimulates angiogenesis [55]. However, FN may be directly produced by the cancer cells as suggested by our multiplex studies. (2) Stimulating extracellular matrix formation: Fibrin forms a scaffold in the TME that can support the adhesion and migration of tumor cells, facilitating their invasion into surrounding tissues. The ECM acts as a barrier to protect cancer from treatments and supports tumor progression [52]. (3) The FN net can act as a reservoir for growth factors and cytokines that support cell growth via a proinflammatory TME. (4) FN can interact with various cells in the microenvironment through signaling pathways. This interaction may support the survival and proliferation of tumor cells or promote their ability to evade the immune system.

Although FN is a component of the ECM, it is often not discussed in papers on ECM or TME for a few reasons: Fibrin is a temporary component of the ECM. Fibrin, unlike collagen or fibronectin, is not associated with the mature tissue structure, but as a somewhat temporary repair stage ECM component [61]. It’s formed during blood clotting in response to injury and wound healing. In the context of tumors, while fibrin may be present due to ongoing vascular permeability or injury, or synthetized locally by the cancer cells, it might not remain in the ECM for extended periods, unlike other ECM components like collagen or fibronectin [62]. Research on the ECM often emphasizes permanent structural components such as collagens, proteoglycans, and glycoproteins. Mainly fibroblasts produce the matrix and regulate matrix remodeling, but in cancer, the tumor matrix also originates from cancer cells [63]. An enriched distribution of fibrin in tumor tissues obtained from high-grade non-small cell lung cancer was demonstrated, and extracellular fibrin was associated with cancer progression, cancer stemness and chemotherapy resistance [64]. Thus, despite its relative under-representation, understanding the role of FN in the ECM is crucial, especially in contexts such as wound healing [65], inflammation, and tumor progression [64], where its presence and transient nature can significantly influence cell behavior and tissue dynamics [66].

#### 4.3.5. Fibrin Deposition-Induced Chronic Inflammation of the Tumor Microenvironment (TME)

Inflammation predisposes to the development of cancer and promotes all stages of tumorigenesis [67]. The TME is a complex and structured mixture characterized by abnormal angiogenesis, chronic inflammation, and dysregulated extracellular matrix (ECM) remodeling, collectively contributing to an immunosuppressive milieu [68]. Chronic inflammation can contribute to cancer progression. Inflammation creates a microenvironment that supports the survival and growth of cancer cells, promotes angiogenesis, and suppresses the immune system’s ability to eliminate cancer cells. Furthermore, an important component of the tumor extracellular matrix is fibrin, which is abundant near tumor vessels [45].

#### 4.3.6. Drugability of Fibrinogen

There are drugs approved and in development targeting fibrinogen for thrombolytic (Abciximab) and non-thrombolytic purposes. Abciximab (ReoPro) inhibits the procoagulant properties (moiety: y400–411) of FN.

As stated above (in the chapter: “The structure of fibrinogen”) the Mac-1-binding site is responsible for the proinflammatory properties of FN. The Mac-1-binding site within the fibrinogen D domain contains two peptide sequences, γ190–202 and γ377–395, designated as P1 and P2. These epitopes differ from the epitopes responsible for the procoagulant function [25]. A FN mutant where only seven amino acids (390–396) are deleted in the inflammatory domain functions under normal hemostasis. For therapeutic application, a first in class antibody (Ab) was demonstrated to recognize only the inflammatory domain of FN—the Ab is targeting a cryptic inflammatory domain of FN exposed only in precipitated fibrin not in soluble fibrinogen [69]. Such an Ab would selectively block the pro-inflammatory function of FN without jeopardizing coagulation. The therapeutic effects have been demonstrated in MS/AD mouse models [41]. The antibody has now been humanized, GMP manufactured and tested in primates, and the first in human studies began in 2023. However, deletion of the minimal recognition sequence, γ390–395, was not sufficient to ablate Mac-1 binding, because mutants were still effective in supporting adhesion of Mac-1-expressing cells probably due to a second sequence in the P1 segment between γ228 and γ253 that contributed to the binding activity [24].

A further possibility of targeted removal of fibrin deposited in tumors would be the targeted delivery of fibrinolytic enzymes to the tumor. In one study tissue plasminogen activator (tPA) was given systemically by intraperitoneal application [45]. However, such a systemic treatment may have bleeding side effects. A more sophisticated application was demonstrated in a recent paper by Ting Mei et al. They demonstrated that redox-active nanoparticles were able to deliver their cargo into solid tumors. Delivery of tPA via these redox-active nanoparticles was able to degrade fibrin in the tumor [44]. A similar approach would be to deliver tPA (Alteplase) or batroxabin (Defibrase) into the tumor via lyso-thermosensitive liposomes (LTSL) and local heating of the tumor [70].

## 5. Summary and Future Perspectives

In conclusion, the presently prevailing concept of the APR may only be part of the truth. There is the well-known systemic acute-phase response, where, in response to tissue injury and consequent inflammation, locally produced inflammatory cytokines (IL-6, IL-1b, etc.) induce acute-phase protein (APP) production by the liver. However, there is also a local APR (lAPR) at the cellular level: a coordinated local production of several acute-phase proteins with the aim of protecting the cells from environmental stress, like the also well-described heat shock response. The HSR exerts its protection within the cell, whereas the lAPR is an extracellular protective mechanism. A future perspective includes the possible utilization of this novel stress response for diagnostic and therapeutic purposes. In cancer, the cancer cells produce APPs to protect themselves from conventional (chemotherapy, radiotherapy) or adjuvant (heat therapy) treatments. Among the identified lAPR proteins fibrinogen appears to be of particular interest as a major component of the TME protecting tumor cells from injury. Local, targeted inhibition of the production of these APPs in the tumor may substantially enhance the anti-tumor effects of currently available anti-cancer treatments. Thus, the potential clinical implications and future directions of research into local acute phase protein production as a therapeutic target in cancer treatment lie primarily in combination therapies: targeting the local acute phase protein production in addition to other therapeutic modalities, such as chemotherapy, radiation therapy, or immunotherapy, may enhance treatment efficacy through synergistic effects. By combining treatments that target different aspects of tumor biology and the immune microenvironment, there lies the potential to overcome resistance mechanisms and improve overall response rates in cancer patients. Furthermore, better understanding of the pathophysiological roles of locally produced APPs may help in the development of targeted immunomodulation: Understanding how APPs are produced and function within the TME could provide insights into ways to modulate local immune responses to better fight cancer. By targeting specific APPs involved in promoting immunosuppression or tumor growth, strategies can be developed to enhance antitumor immune response and improve the efficacy of cancer immunotherapy.

## Figures and Tables

**Figure 1 ijms-25-03435-f001:**
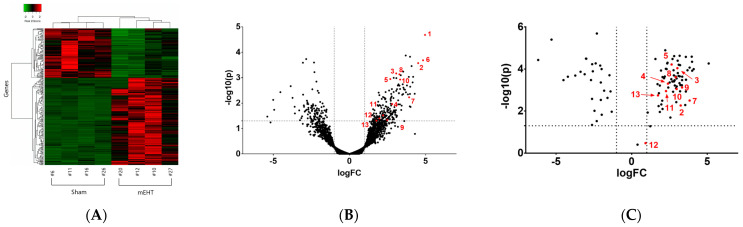
Multiplex analysis of mEHT treated TNBC samples. (**A**) Next generation sequencing (NGS) results of mEHT- vs. sham-treated tumors (heat map: red: up-, green: down-regulated genes). (**B**) Volcano plot of the NGS data: the most upregulated proteins were acute phase proteins (red dots and numbers). (**C**) NanoString verified acute phase protein upregulation [4].

## Data Availability

Not applicable.

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
