# Peer review of "Local Production of Acute Phase Proteins: A Defense Reaction of Cancer Cells to Injury with Focus on Fibrinogen"

_ijms, 2024, doi:10.3390/ijms25063435_

Round 1
Reviewer 1 Report
Comments and Suggestions for Authors
Briefly, the review deals with collected data on the treatment of triple-negative breast cancer with modulated electro-hyperthermia in a mouse model, with emphasize on fibrinogen roles. Also, the author's team (according to the acknowledgment and published articles in References) propose that the local acute phase response is a new, novel cellular defense mechanism like the heat shock response.
Although it is evident that the author has vast experience and previously published articles in this field, which generally support the performed experiments presented also in this paper, I have to notice that this manuscript was sometimes pretty difficult to follow. The manuscript body was written almost as a draft version in certain parts. The sections are not labeled, except Section 5, so the reader will find it difficult to know which part he is reading.
We find the Introduction and Background, which is also confusing. It would be better to mark the headings with numbers and sub-numbers if they belong to the part you wish to write about. Some paragraphs you start with bold letters, some titles contain different fonts, you use ’’italic’’ for the word liver and many other inflammatory conditions in the Section or Subsection (line 163). Why is that? There are also bold letters for IBD (line 176)? The font differs throughout the text, justifying also. Line 154-there is a bracket... These are examples for the impression this is a draft version of the manuscript.
Comments on the Quality of English LanguageEnglish language needs minor corrections
Author Response
"Briefly, the review deals with collected data on the treatment of triple-negative breast cancer with modulated electro-hyperthermia in a mouse model, with emphasize on fibrinogen roles. Also, the author's team (according to the acknowledgment and published articles in References) propose that the local acute phase response is a new, novel cellular defense mechanism like the heat shock response."
Answer: Thank you for reviewing this paper, for your understanding of the message and for your valuable comments improving the manuscript.
"Although it is evident that the author has vast experience and previously published articles in this field, which generally support the performed experiments presented also in this paper, I have to notice that this manuscript was sometimes pretty difficult to follow. The manuscript body was written almost as a draft version in certain parts. The sections are not labeled, except Section 5, so the reader will find it difficult to know which part he is reading."
Answer: Unfortunately, the text the referee reffers to is the uploaded word version of the paper, not the uploaded pdf in which all sections were labeled. The reviesed MS has numbered chapters.
"We find the Introduction and Background, which is also confusing."
Answer: The introduction is a general introduction of the topic, whereas the background summarizes main characteristics of the model used in the reviewed experiments. The titles are more specific now in the revised manuscript.
"It would be better to mark the headings with numbers and sub-numbers if they belong to the part you wish to write about."
Answer: see above: The reviesed MS has numbered chapters now.
"Some paragraphs you start with bold letters, some titles contain different fonts, you use ’’italic’’ for the word liver and many other inflammatory conditions in the Section or Subsection (line 163). Why is that? There are also bold letters for IBD (line 176)? The font differs throughout the text, justifying also. Line 154-there is a bracket... These are examples for the impression this is a draft version of the manuscript."
Answer: Thank you for pointing out these mistakes. I have corrected them: unified the font and the justifying and removed unnecessary bold and italic formattings. I also formatted titles uniformly with numbering and subnumbering and unifrom: bold, calibri, 12 pt font.
Reviewer 2 Report
Comments and Suggestions for Authors
The authors have reviewed the papers and described current advances associated with the roles of fibrinogen in treatment of triple-negative breast cancer (TNBC) with modulated electro-hyperthermia (mEHT). The manuscript raises several interesting points of view concerning the application of fibrinogen as potential new therapeutic options in cancer. However, there are still several important issues should be mentioned in this manuscript.
1. First of all, the Fig. 2A&B have been published in the previous manuscripts. It is just a pale sequel of the previous report. Lack of novel points of view in this field.
2. There is no systematic organization in the molecular mechanisms associated with the connections between fibrinogen and tumor progression.
3. The authors should provide easily read tables for functional roles of fibrinogen and defense reaction of tumor tissues to enhance the value of this manuscript.
4. Lack of Fig. 1 in the context.
5. Some of the sentences are disorganized and difficult to read; the phrases as well as grammar errors in the manuscript require further reorganization and corrections.
Comments on the Quality of English Language
Extensive editing of English language required.
Author Response
"The authors have reviewed the papers and described current advances associated with the roles of fibrinogen in treatment of triple-negative breast cancer (TNBC) with modulated electro-hyperthermia (mEHT). The manuscript raises several interesting points of view concerning the application of fibrinogen as potential new therapeutic options in cancer. However, there are still several important issues should be mentioned in this manuscript."
Answer: Thank you for reviewing this paper and for your valuable comments improving the manuscript.
- "First of all, the Fig. 2A&B have been published in the previous manuscripts. It is just a pale sequel of the previous report. Lack of novel points of view in this field."
Answer: Indeed these 2 figures are repetitions from the previuos paper. I repeated them, to illastrate the claims regarding HSP70 staining. They are deleted from the revised manuscript as HSP70 is not a central part of the MS.
- "There is no systematic organizationin the molecular mechanisms associated with the connections between fibrinogen and tumor progression. 3. The authors should provide easily read tables for functional roles of fibrinogen and defense reaction of tumor tissues to enhance the value of this manuscript."
Answer: Table 1 summarizing FN function/molecular mechanism, FN’s role in cancer progression, supporting experimental evidence have been added to the revised MS.
- "Lack of Fig. 1 in the context."
Answer: Indeed fig. 1 and 2 were merged and thus fig. 1. was deleted in the pre-final version. The figures are renumbered correctly in the revised MS.
- "Some of the sentences are disorganized and difficult to read; the phrases as well as grammar errors in the manuscript require further reorganization and corrections."
Answer: A native speaker college has corrected the manuscript.
Reviewer 3 Report
Comments and Suggestions for Authors
The authors start the abstract with the purpose of the study, not with the results obtained. The abstract is disorganized and not clear.
The introduction presents very few data from the specialized literature and also very few references.
After all, what is this article? Original or review type?
I consider that the authors should redo the structure of this article.
Author Response
"The authors start the abstract with the purpose of the study, not with the results obtained. The abstract is disorganized and not clear."
Answer: Indeed the abstract was not clear, so it has been rewritten in the revised manuscript.
"The introduction presents very few data from the specialized literature and also very few references."
Answer: As this is a review paper, the introduction and the background chapters give the background for the paper. These chapters are relabeled in the revised MS for better clarity.
"After all, what is this article? Original or review type?"
Answer: this is a review article, however it is based on observations from our own experiments which are compared with and supported by the literature.
"I consider that the authors should redo the structure of this article."
Answer: The paper has been restructured including numbering and subnumbering of the headings, extending the text at several parts, renumbering the figures, etc.
Round 2
Reviewer 1 Report
Comments and Suggestions for Authors
The manuscript is corrected and is acceptable in the present form.
Author Response
Thank you for your review; we are pleased to hear that the manuscript meets your approval.
Reviewer 2 Report
Comments and Suggestions for Authors
1. Figure 1 or 2 is still unclear, it is easy to confuse for author, such as line 114 list figure 2/E???
2. Fibrin vs ECM in Figure 2 or 3, the author cannot connect these two factors at the same figure?
Comments on the Quality of English LanguageOK
Author Response
"Figure 1 or 2 is still unclear, it is easy to confuse for author, such as line 114 list figure 2/E???"
Answer:
I removed the reference to Fig. 1 from line 61.
I modified the reference to Fig. 2/C-E to Fig.2 in line 92.
I modified the reference to Fig. 2/E to Fig.1/C in line 101.
I removed the reference to Fig. 2 from line 218.
Thank you for drawing my attention to these still present mistakes.
"Fibrin vs ECM in Figure 2 or 3, the author cannot connect these two factors at the same figure?"
Answer: It is now Fig. 2 in the revised MS (there is no fig. 3). Panel A is about the role of fibrin in cancer progression. The abbreviations: ECM/BM have been removed from the revised MS.
I appreciate your careful work on this manuscript.
Reviewer 3 Report
Comments and Suggestions for Authors
The article can be published.
Author Response
Thank you for your review. We are pleased to hear that the manuscript meets your approval.
Round 3
Reviewer 2 Report
Comments and Suggestions for Authors
Substantial corrections by the author may be accepted for publication.
Comments on the Quality of English LanguageOK